# Manganese Oxide Nanoparticles as Safer Seed Priming Agent to Improve Chlorophyll and Antioxidant Profiles in Watermelon Seedlings

**DOI:** 10.3390/nano11041016

**Published:** 2021-04-15

**Authors:** Deepak M. Kasote, Jisun H. J. Lee, Guddarangavvanahally K. Jayaprakasha, Bhimanagouda S. Patil

**Affiliations:** Vegetable and Fruit Improvement Center, Department of Horticultural Sciences, Texas A&M University, 1500 Research Parkway, Suite A120, College Station, TX 77845-2119, USA; deepakkasote06@gmail.com (D.M.K.); jslee@tamu.edu (J.H.J.L.); gkjp@tamu.edu (G.K.J.)

**Keywords:** antioxidant activity, hormone, manganese oxide nanoparticles, metabolomics, chlorophylls, watermelon seedlings

## Abstract

The use of nanoscale nutrients in agriculture to improve crop productivity has grown in recent years. However, the bioefficacy, safety, and environmental toxicity of nanoparticles are not fully understood. Herein, we used onion bulb extract to synthesize manganese oxide nanoparticles (MnO-NPs). X-ray diffraction, X-ray photoelectron spectroscopy, and high-resolution transmission electron microscopy were used for the structural and morphological characterization of synthesized MnO-NPs. The MnO-NPs were oval shape crystalline nanoparticles of Mn_2_O_3_ with sizes 22–39 nm. In further studies, we assessed the comparative toxicity of seed priming with MnO-NPs and its bulk counterparts (KMnO_4_ and Mn_2_O_3_), which showed seed priming with MnO-NPs had comparatively less phytotoxicity. Investigating the effect of seed priming with different concentrations of MnO-NPs on the hormonal, phenolic acid, chlorophyll, and antioxidant profiles of watermelon seedlings showed that treatment with 20 mg·L^−1^ MnO-NPs altered the chlorophyll and antioxidant profiles of seedlings. At ≤40 mg·L^−1^, MnO-NPs had a remarkable effect on the phenolic acid and phytohormone profiles of the watermelon seedlings. The physiological outcomes of the MnO-NP seed priming in watermelon were genotype-specific and concentration-dependent. In conclusion, the MnO-NPs were safer than their bulk counterparts and could increase crop productivity.

## 1. Introduction

The global agricultural output needs to increase by an estimated 60% by 2050 to assure food security for the predicted population of 9 billion people [1,2]. The use of innovative technologies such as nanotechnology in agriculture could be crucial to make agriculture more productive and sustainable [3]. Nanosensor systems, nanoscale nutrients, and pesticides are gaining popularity in agriculture [4,5]. However, concerns are also rising in the scientific community about the use of nanoscale nutrients and pesticides; these concerns are mainly related to their bioefficacy, biosafety, and environmental toxicity. Hence, recent research has been tilted towards developing eco-friendly nanomaterials [6].

Biotic and abiotic stresses cause considerable and unpredictable losses to crops [7]. Several methodologies, such as conventional breeding, mutation breeding, polyploidy breeding, genetic engineering, and seed priming, have been adapted to accelerate seedling emergence in the field and to impart tolerance to plants against adverse conditions [8]. Various seed priming (presoaking) techniques such as priming with water, salts, osmoticum, solid matrices, different chemicals, temperature, and plant hormones have been developed [8]. Amongst these, seed priming in nanoparticle solutions (often termed “nanopriming”) is gaining importance in crop science. Nanopriming is reported to improve seed germination, growth, and yield of crops [9,10]. Thus far, several biogenic and synthetic metallic nanoparticles and carbon nanotube-based seed priming methods have been used to improve seed germination and plant growth [10,11,12,13].

Plants require micronutrient manganese (Mn) for photosynthesis, respiration, and other processes. The Earth’s crust is rich in Mn [14,15], which is easily oxidized to produce over 30 manganese oxides (MnO_x_)/hydroxide [14]. Compare to the conventional bulk or ionic Mn compounds, nanoscale Mn is found to be less phytotoxic and more effective in minimizing abiotic stresses in plants [16]. MnO_x_ nanoparticles are mainly produced by bacteria in the environment [15]. In recent studies, phytoextracts have been used in the production of Mn nanoparticles [17,18]. However, the physiological and toxicological effect of the various Mn nanoparticles on agricultural crops is relatively less known.

In this study, the synthesis of manganese oxide nanoparticles (MnO-NPs) was carried out using onion bulb extract. X-ray diffraction (XRD), X-ray photoelectron spectroscopy (XPS), and high-resolution transmission electron microscopy (HR-TEM) techniques were used for the structural and morphological characterization of MnO-NPs. In further studies, the comparative toxicity of seed priming with MnO-NPs and its bulk counterparts (KMnO_4_ and Mn_2_O_3_) was studied. The effect of MnO-NPs seed priming treatments of different concentrations on the hormonal, phenolic acid, chlorophyll, and antioxidant profiles of watermelon seedlings was also investigated to understand the nature of the interactions between MnO-NPs and plants. Moreover, we used the green approach to synthesize MnO-NPs and showed they are safer than their bulk counterparts; this information can also help to synthesize ecofriendly MnO-NPs for their non-agri-food applications, such as in biofuel and batteries, where the ecotoxicity of the nanomaterials is the primary concern.

## 2. Materials and Methods

### 2.1. Materials and Chemicals

Waste onion bulbs were obtained from local growers and supermarkets. The seeds of diploid (Riverside) and triploid (Maxima) watermelon varieties obtained from Origene Seeds Ltd., Giv’aat Brener, Israel. Abscisic acid (ABA), 2,2-diphenyl-1-picrylhydrazyl (DPPH), gibberellic acid (GA), jasmonic acid (JA), potassium permanganate (KMnO_4_), salicylic acid (SA), zeatin (ZA), and phenolic acids (4-hydroxy-benzoic acid, caffeic acid, phthalic acid, protocatechuic acid, and *trans*-cinnamic acid) were procured from Sigma Aldrich (St. Louis, MO, USA). 12-oxo phytodienoic acid (OPDA) purchased from Cayman Chemical, Ann Arbor, MI, USA. 2,2′-azino-bis(3-ethylbenzothiazoline-6-sulfonic acid (ABTS) was obtained from Chem-Impex Int’l. Inc. (Bensenville, IL, USA).

### 2.2. Synthesis of MnO-NPs

For the synthesis of MnO-NPs, the onion bulb extract was prepared by grinding sliced onion bulbs, as per the procedure described previously [19]. Then, the onion extract was slowly added to 0.01 M KMnO_4_ solution (1:1, *v*/*v*) and kept on a magnetic stirrer with constant stirring for two h at ambient temperature (pH 7). The resultant black precipitate was separated by centrifuging at 7741× *g* for 10 min. The residue was washed with nanopure water, and the aqueous phase was removed after centrifuging at 7741× *g* for 10 min. This procedure was repeated three times, and finally, the residue was washed with absolute ethanol. After freeze-drying for 14 h (Labconco Freeze Dryer System, Kansas City, MO, USA), the residue was calcined for 4 h at 500 °C in a muffle furnace (Thermo Scientific, Pittsburgh, PA, USA). The final black-colored fine MnO-NP powder was stored in a cool and dry place until its use for further characterization and nanopriming studies.

### 2.3. Characterization of MnO-NPs

The XRD, XPS, and HR-TEM analyses of dried MnO-NP powder were carried out as per procedures described in our previous published study [20].

### 2.4. Phytosafety of MnO-NP Seed Priming 

For seed priming, three different concentrations (25, 50, and 100 mg·L^−1^) of MnO-NPs, bulk KMnO_4,_ and Mn_2_O_3_ solutions were prepared in nanopure water. Under each treatment, diploid watermelon seeds (*n* = 20, in triplicate) were presoaked in 15 mL of priming solution for 14 h. Seeds soaked in nanopure water were used as a hydro-primed treatment. The next day, after drying, unprimed and primed seeds were sown in a plug tray for seedling development [20]. On the eighth day, leaf samples from seedlings of each group were used for leaf disc-based antioxidant activity measurement assays and chlorophyll estimation.

### 2.5. Nanopriming, Seed Germination, and Seedling Development

In the nanopriming procedure, 20 watermelon seeds (diploid and triploid varieties) were primed in 15 mL of aqueous MnO-NP suspension (10, 20, 40, and 80 mg·L^−1^) for 14 h. For the germination test, unprimed, hydroprimed, and MnO-NP-primed seeds were placed on Petri plates containing a sterile filter paper. First, the filter paper was wetted with 10 mL nanopure water, and around 8 mL water was added to each plate on alternate days [20]. The germination tests were carried out in an incubator at 28 °C under dark conditions. The germinated seeds were counted daily, and the final results of germination tests were expressed as the mean number of days to 50% germination [21].

For seedling development, another set of unprimed, hydroprimed, and MnO-NP-primed seeds (20 seeds in triplicate under each treatment) were sown in 200-cell trays. Trays were then transferred to the growth chamber for seedling development for eight days. After that, seedlings from each treatment group were collected, and root and stem lengths were measured. Fresh leaf samples from each treatment group were used for leaf disc-based antioxidant activity measurement assays. The rest of the leaf tissue samples were stored at −80 °C for further untargeted plant metabolomics study and quantitative profiling of chlorophylls, phytohormones, and phenolic acids [20].

### 2.6. Antioxidant Activity and Chlorophylls Measurement

The antioxidant activity profiles of unprimed, hydroprimed, and MnO-NP-primed diploid and triploid watermelon seedling samples were studied using novel ABTS, DPPH, and potassium permanganate reduction (PPR) leaf-disc assays developed in our lab [19]. The chlorophyll extraction was conducted as per the procedure described in the previous study [20], and estimation was carried out by measuring absorbance at 652 and 665 nm in a microplate reader. The following equations were used to estimate the contents of chlorophyll *a* and chlorophyll *b* per gram of fresh weight (FW) [22,23].
Chlorophyll *a* (µg·g^−1^ FW) = (16.72 × A_665_ − 9.16 × A_652_) × 50
Chlorophyll *b* (µg·g^−1^ FW) = (34.09 × A_652_ − 15.28 × A_665_) × 50
where A_652_ and A_665_ are the absorbances at 652 and 665 nm, respectively.

### 2.7. Untargeted Metabolomics and Phenolic Acids Profiling

The sample preparation and UHPLC/ESI-HR-QTOFMS based quantitative profiling of phenolic acids were performed as per our previously described methods [19,20]. For untargeted metabolomics study, the UHPLC/ESI-HR-QTOFMS data were preprocessed by DataAnalysis 4.3 Software (Bruker Daltonics, Bremen, Germany), and multivariate analysis was performed using online software MetaboAnalyst 3.0 [20,24].

### 2.8. Hormone Analysis

The phytohormones were extracted as per the protocol described in our previous study [20]. UHPLC/ESI-HR-QTOFMS was used for the quantitative profiling of hormones [20].

### 2.9. Statistical Analysis

Results were expressed as a mean ± standard error (SE). Statistical analysis was per-formed using the SPSS software (Version 25, IBM SPSS Statistics, IBM Corp., Chicago, IL, USA).

## 3. Results

### 3.1. Synthesis and Characterization of MnO-NPs

The production of nanoscale nutrients using a green approach can be an effective strategy to minimize concerns with their biosafety and environmental toxicity. Considering this, in the present study, the green synthesis of MnO-NPs was carried out using onion extract. The clear purple KMnO_4_ solution immediately changed to a blackish-brown colloidal solution after adding onion extract, which demonstrated the formation of MnO-NPs. The phytochemicals, mainly polyphenols from the onion extract, may reduce permanganate (MnO_4_^−^) into manganese dioxide (MnO_2_) by transferring three electrons. The organic matter from synthesized MnO-NPs was further removed by calcination. Heating MnO-NPs at high temperatures (500 °C) in the air may lead to the formation of more crystalline MnO-NPs.

Figure 1a shows the XRD pattern of synthesized MnO-NPs. The sharp diffraction peaks confirmed the crystalline nature of MnO-NPs. In the XRD spectrum, the observed diffraction peaks at 2θ values of 23.30°, 33.07°, 38.42°, 45.31°, 49.46°, 55.22°, and 65.88° correspond to crystal planes (211), (222), (400), (332), (431), (440), and (622), respectively, which demonstrates the formation of Mn_2_O_3_ [25]. The results of TEM analysis showed synthesized MnO-NPs had oval shapes of 22–39 nm (Figure 1b). The magnified HRTEM image also demonstrated a lattice fringe spacing of 0.27 nm, which corresponds to the (222) plane of Mn_2_O_3_ (Figure 1c) [26].

The surface electronic state of Mn_2_O_3_ was also confirmed by XPS analysis. Figure 1d shows the typical survey spectrum of MnO-NPs, containing photoelectron lines of phosphorus, carbon, calcium, potassium, oxygen, and manganese. The narrow scan XPS spectrum of the Mn 2p-electrons showed two photoelectron lines at approximately 641.4 eV and 653.5 eV, which can be attributed to Mn 2p3/2 and Mn 2p1/2 levels, respectively (Figure 1e). These binding energies agreed well with the reported Mn_2_O_3_ values [27]. The high-resolution O 1s XPS spectrum exhibits a strong peak at 530.9 eV, which corresponds to the lattice O in Mn_2_O_3_ (Figure 1f) [28].

### 3.2. Comparative Phytotoxicity of MnO-NPs, Bulk KMnO_4,_ and Mn_2_O_3_

As of now, the use of nanoscale nutrients in agriculture is limited due to the lack of information about their toxicity in crops. It has been reported that conventional bulk/ionic Mn compounds can induce abiotic stress in plants [16]. Figure 2 shows the comparative results of the antioxidant activity and chlorophyll contents in leaf samples from different seed priming treatments with MnO-NPs, bulk KMnO_4_, and Mn_2_O_3_. The studied MnO-NP seed priming treatments had no considerable impact on chlorophyll contents (Figure 2a). However, seed priming with its bulk counterpart, Mn_2_O_3,_ significantly reduced the chlorophyll *a* and *b* contents in diploid watermelon seedlings (Figure 2a). Conversely, seed priming with KMnO_4_ up to 50 mg·L^−1^ had no influence on chlorophyll *a* and *b* contents.

Mn is an essential micronutrient for plants. It is mainly involved in photosynthesis as an enzyme cofactor in the water-splitting reaction of photosystem II (PSII), and also helps to lower the reactive oxygen species (ROS) [29,30,31,32]. However, excess Mn can be toxic, leading to the inhibition of chlorophyll synthesis [33]. Similar to our findings, MnO-NPs showed less toxicity than their ionic counterparts in lettuce (*Lactuca sativa* L.) [33]. Moreover, Mn nanoparticles increased photophosphorylation, water splitting by an oxygen-evolving complex, and nitrogen assimilation in plants compared with its elemental or bulk counterparts [34,35].

In plants, reactive oxygen species (ROS) production is sensitive to various metallic nanoparticles [36,37]. The changes in ROS production further affect the enzymatic and nonenzymatic antioxidant defenses in plants [38]. In plants, Mn toxicity is mitigated by activating antioxidant defenses. Figure 2b indicates the levels of antioxidant activities in the leaf samples of diploid watermelon seedlings after seed priming treatments with MnO-NPs, KMnO_4,_ and Mn_2_O_3_ in comparison with controls. Seed priming with different concentrations of MnO-NPs had no considerable effect on the observed radical scavenging activities in DPPH and ABTS leaf disc assays. However, the antioxidant activity values in the ABTS assay were significantly increased in KMnO_4_ and Mn_2_O_3_ seed priming treatments at concentrations ≤ 50 mg·L^−1^. Interestingly, we found that the reducing power values in PPR leaf assays significantly decreased after MnO-NPs, KMnO_4,_ and Mn_2_O_3_ seed priming treatments (Figure 2b). These results collectively indicate that MnO-NPs have less phytotoxicity than their bulk counterparts (KMnO_4_ and Mn_2_O_3_).

### 3.3. Effect of Different MnO-NPs Seed Priming Treatments on Seed Germination, Seedling Growth Parameters, Photosynthetic Pigments, and Antioxidant Potential

Seed priming is a promising approach to improve germination and seedling development by altering the physiological state of the seed [7,8]. Recently, the technique of seed priming with synthetic nanoparticles, typically called “nanopriming” has also been gaining importance in improving desired traits in crops [9]. Moreover, nanopriming is a controlled technique, which minimizes the exposure risk of nanoparticles in the environment. Figure 3a shows the results of mean days to 50% germination of unprimed, hydroprimed, and different MnO-NPs seed priming treatments in two watermelon varieties. The hydropriming and 10 mg·L^−1^ MnO-NPs seed priming treatments significantly enhanced the seed germination compared to unprimed control in the diploid genotype. However, the above positive effect was not observed in the triploid variety. Seed priming with MnO-NPs at concentrations ≤ 20 mg·L^−1^ nonsignificantly reduced seed germination in the diploid watermelon variety (Figure 3a). A previous study reported that MnO-NPs had no considerable impact on seed germination [33]. However, they improved the seedling growth in lettuce and increased root growth in *Capsicum annuum* L. [33,39]. We also investigated the effects of the different MnO-NPs priming treatments on seedling growth in the diploid and triploid varieties (Appendix A). All the investigated MnO-NPs priming treatments had no significant influence on the stem and root lengths. These findings confirmed that seed priming with MnO-NPs had no considerable phytotoxicity up to the studied concentration of 80 mg·L^−1^ in standard phytotoxicity tests such as seed germination and root elongation.

Chlorophyll *a* is a primary pigment, along with other accessory pigments (chlorophyll *b*), in harvesting light energy [40]. We investigated the effect of MnO-NPs priming treatments at different concentrations on the synthesis of chlorophyll *a* and *b* in the diploid and triploid watermelon varieties (Figure 3b). Indeed, MnO-NP priming treatments dose-dependently affected the chlorophyll *a* and *b* levels in triploid watermelon seedlings. At 20 mg·L^−1^ MnO-NPs, chlorophyll *a* and *b* levels were significantly increased in triploid watermelon seedlings compared with unprimed and hydroprimed controls. However, in the triploid variety, MnO-NPs priming treatments below and above 20 mg·L^−1^ did not influence the chlorophyll *a* and *b* levels. On the other hand, the chlorophyll *a* and *b* levels in diploid watermelon seedlings were nonsignificantly altered in response to different MnO-NPs priming treatments, demonstrating that chlorophyll synthesis in watermelon seedlings may be highly sensitive to the concentration of Mn. The genetic makeup and seed morphology may be responsible for the different chlorophyll *a* and *b* profiles in the diploid and triploid varieties [20]. The outcome of any seed priming treatment is generally influenced by the plant genotype, seed morphology, and physiology. The uptake of nanoparticles in the various nanopriming treatments is found to differ, and hence their physiological outcomes in plants vary with size and unique physicochemical properties [41,42].

Figure 3c shows the antioxidant activity levels in DPPH, ABTS, and PPR leaf disc assays after different MnO-NP and hydropriming treatments, compared with untreated control. The MnO-NPs treatments had a differential impact on the antioxidant activities of nonenzymatic antioxidants in diploid and triploid watermelon seedlings. In diploid watermelon seedlings, nonsignificant changes were observed in the antioxidant activity levels in DPPH and DPPH leaf assays after all MnO-NPs treatments compared with control plants. However, in the triploid variety, at higher MnO-NPs concentrations (80 mg·L^−1^), antioxidant activity levels in DPPH leaf disc assay were increased. Conversely, at higher MnO-NPs concentrations (≤40 ppm), the values of reducing power in diploid and triploid watermelon seedlings leaf samples were significantly reduced in PPR assay compared with unprimed control. These findings indicate that a higher dose of MnO-NPs may exert oxidative stress in watermelon plants. The literature also reported that the MnO-NPs produced oxidative stress and toxic effects at higher concentrations (50–200 mg·L^−1^) in deadly nightshade (*Atropa belladonna* L.) [43].

### 3.4. Untargeted Metabolomics to Understand the Influence of Priming with Different Concentrations of MnO-NPs on the Watermelon Leaf Metabolome

Metabolomics can be an effective approach to understand the nature of interactions between nanomaterials and plants. Herein, principal component analysis (PCA) and partial least squares discriminant analysis (PLS-DA) multivariate analysis methods were used to explore the impacts of the different MnO-NPs treatments on the leaf metabolite responses in diploid and triploid watermelon seedlings. In the PCA scores plots, a clear separation was observed between the unprimed control and primed samples (Figure 4a,b). However, the separation between the MnO-NPs priming treatments and the hydroprimed control was only clearly found in the samples of the triploid watermelon variety (Figure 4b).

Supervised PLS-DA was further performed to sharpen the variability among the metabolomes of the hydroprimed and different MnO-NPs-primed samples. As shown in Figure 4c,d, distinct clusters were found for unprimed, hydroprimed, and different MnO-NPs-primed samples in PLS-DA scores plots of diploid and triploid varieties. These findings demonstrated that seed priming with MnO-NPs modulates metabolites in watermelon plants according to their treatment concentrations.

### 3.5. Seed Priming with MnO-NPs Altered Phytohormone Profiles Distinctly in Diploid and Triploid Watermelon Varieties

Plant hormones regulate the growth and development of plants, and they have a crucial role in the responses of the plants to different stresses. The assessment of the plant hormones pool provides an overview of the interaction of the plants with external factors. In our previous study, plant hormones were affected in watermelon seedlings in response to iron nanoparticle treatments [20]. The influence of the different MnO-NPs priming treatments on leaf phytohormone profiles in diploid and triploid watermelon seedlings is shown in Figure 5. The studied MnO-NPs priming treatments distantly modulate phytohormone profiles in diploid and triploid varieties of watermelon. Similar to our results, Iqbal and coworkers also found that different priming agents (i.e., CaCl_2_, KCl, and NaCl) distinctly influenced phytohormone profiles of two wheat cultivars [44]. We further found that the changes in the phytohormone profiles in diploid and triploid varieties were MnO-NPs concentration-dependent.

In diploid seedlings, MnO-NPs priming treatments significantly modulated the levels of ABA, JA, and OPDA compared with unprimed and hydroprimed controls. Seed priming with 20 to 80 mg·L^−1^ MnO-NPs significantly reduced the level of OPDA, and conversely, increased the level of JA, demonstrating MnO-NP priming promotes conversion of OPDA into JA in diploid watermelon seedlings. Previous studies showed that the lowering of OPDA accumulation might be helpful to break seed dormancy and improve plant development [45]. However, this was not observed in the triploid watermelon variety. In our previous study, seed priming with iron oxide nanoparticles also increased the conversion of OPDA into JA in diploid watermelon seedlings [20].

The observed phyto-ormone profiles of triploid watermelon seedlings in response to different MnO-NPs priming treatments were distinct from those measured in the leaf samples of diploid seedlings (Figure 5). The levels of ABA, GA, and JA were significantly altered in response to different MnO-NP priming treatments in triploid seedlings. In triploid seedlings, priming with 40 and 80 mg·L^−1^ of MnO-NPs significantly increased levels of ABA and GA, whereas this treatment significantly reduced SA and ZA levels. Among these hormones, ABA is a stress-responsive hormone [46]. The accumulation of ABA in triploid seedlings at higher priming concentrations of MnO-NPs demonstrates that the watermelon plant experienced stress at the higher seed priming concentration.

### 3.6. Seed Priming with Higher Concentrations of MnO-NPs Modulates Leaf Phenolic Acid Profiles in the Watermelon Plant

Plants trigger the synthesis of phenolic compounds upon exposure to nanoparticles as a protective response against oxidative stress [20]. In this study, the levels of plant phenolics in the watermelon seedling were measured to understand the protective response of the plant to different treatments of MnO-NPs (Table 1). The MnO-NP priming treatments up to concentrations of 40 mg·L^−1^ had no significant impact on the studied phenolic acid profiles of diploid and triploid watermelon seedlings (Table 1). However, at higher seed priming concentrations (80 mg·L^−1^), MnO-NPs significantly increased the accumulation of *trans*-cinnamic acid in the leaf tissue of both diploid and triploid varieties (Table 1). It has been reported that *trans*-cinnamic acid acts as autotoxin, and it was found to trigger oxidative stress in cucumber plants [47]. Conversely, the observed protocatechuic acid content was reduced to below the detectable limit in diploid and triploid seedlings at seed priming concentration, 80 mg·L^−1^ (Table 1). Protocatechuic acid is known to have high chelating strength. In our study, a below-detectable level of protocatechuic acid at higher concentrations of MnO-NPs may indicate its chelation with MnO-NPs.

## 4. Conclusions

In this study, we have shown that onion extract can be efficiently used for the production of oval crystalline Mn_2_O_3_ nanoparticles (MnO-NPs) of 22–39 nm. Seed priming with MnO-NPs had less phytotoxicity than the bulk form of Mn present in KMnO_4_ and Mn_2_O_3_. This study showed that MnO-NPs considerably affect the chlorophyll and antioxidant profiles at 20 mg·L^−1^. Conversely, at ≤40 mg·L^−1^ the priming had a remarkable influence on leaf phenolic acid and phytohormone profiles in watermelon seedlings. However, the observed physiological outcomes of MnO-NPs seed priming treatments in watermelon were genotype-specific and concentration-dependent. The results of the present study collectively demonstrated that green synthesized MnO-NPs could be a safer seed priming agent compared to bulk counterparts, KMnO_4_ and Mn_2_O_3_ to improve the productivity of the watermelon crop at its optimal effective concentration. However, further studies regarding the exact influence of seed priming with MnO-NPs on overall agricultural output, including its role in providing tolerance against various abiotic and biotic stresses to other horticultural crops, are warranted.

## Figures and Tables

**Figure 1 nanomaterials-11-01016-f001:**
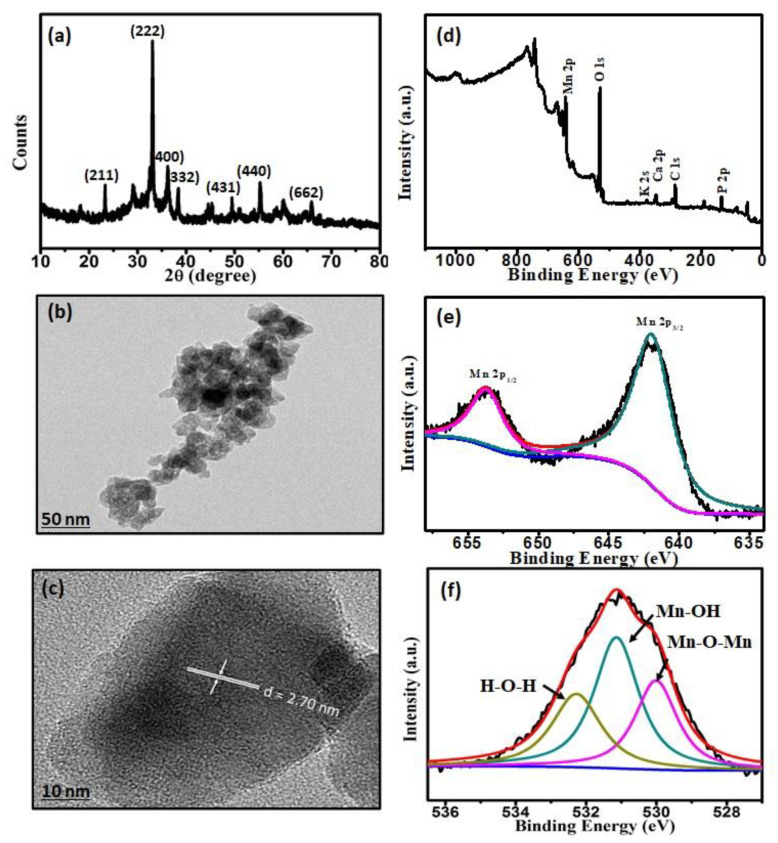
Characterization of MnO-NPs: (**a**) X-ray diffraction (XRD) pattern; (**b**) transmission electron microscopy (TEM) image; (**c**) high resolution (HR)-TEM image; (**d**) X-ray photoelectron spectroscopy (XPS) overview spectrum of MnO-NPs; (**e**,**f**) the narrow scan Mn 2p-electrons and O1s XPS spectrum of MnO-NPs, respectively.

**Figure 2 nanomaterials-11-01016-f002:**
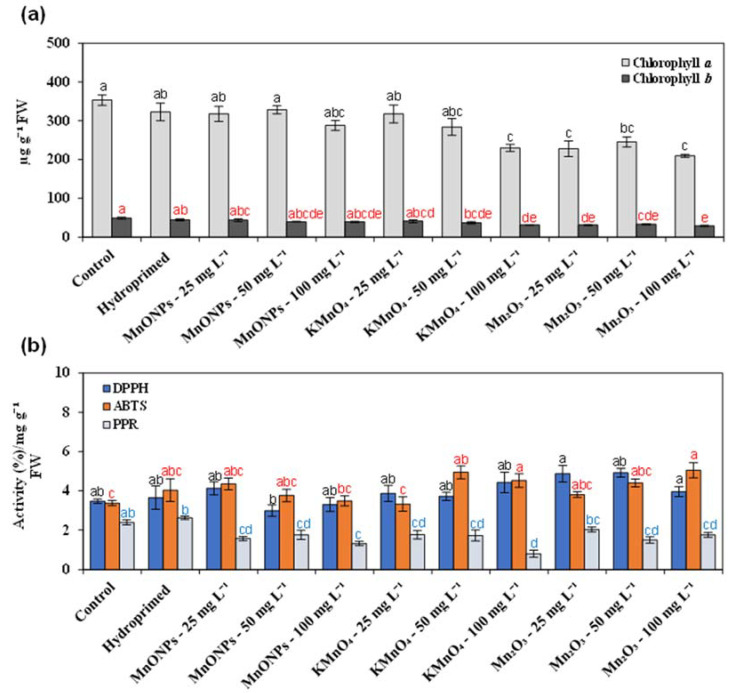
Comparative phytotoxicity of different seed priming treatments with MnO-NPs, KMnO_4,_ and Mn_2_O_3_ in diploid watermelon seedlings. The effect of MnO-NPs, KMnO_4,_ and Mn_2_O_3_ seed priming treatments (25, 50, and 100 mg·L^−1^) on chlorophyll contents (**a**) and antioxidant activity levels in DPPH, ABTS, and PPR leaf disc assays (**b**). The post hoc test based significant differences (*p* < 0.05) among different groups, and assays are presented by different letters and colors.

**Figure 3 nanomaterials-11-01016-f003:**
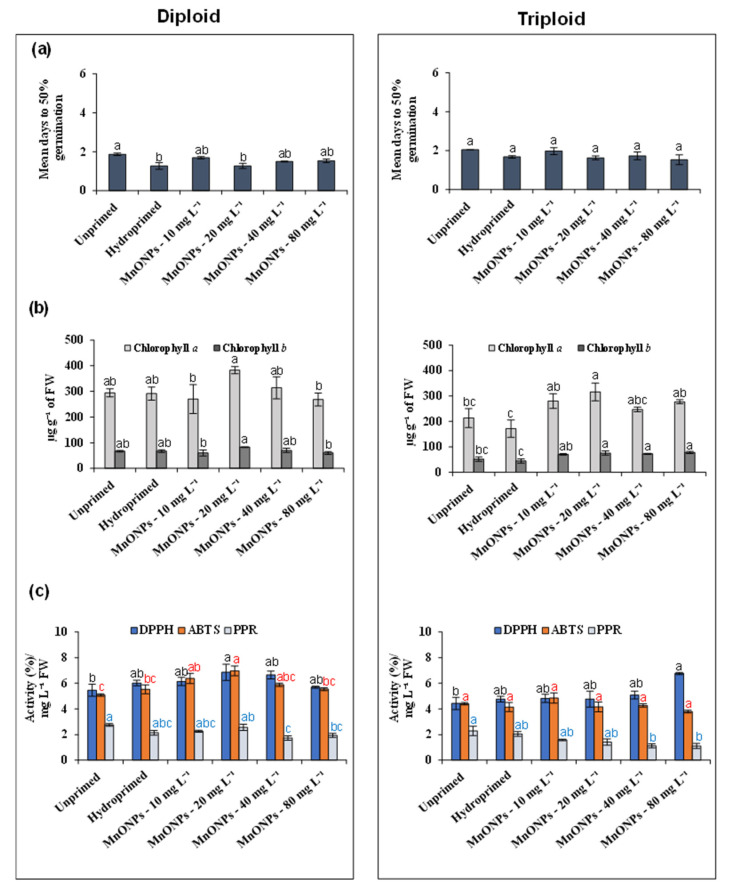
Effect of different MnO-NPs seed priming treatments on mean days to reach 50% germination (**a**), chlorophyll contents (**b**), and antioxidant activities (**c**) in diploid and triploid watermelon seedlings. The post hoc test significant based differences (*p* < 0.05) among different groups and assays are shown by different letters and colors.

**Figure 4 nanomaterials-11-01016-f004:**
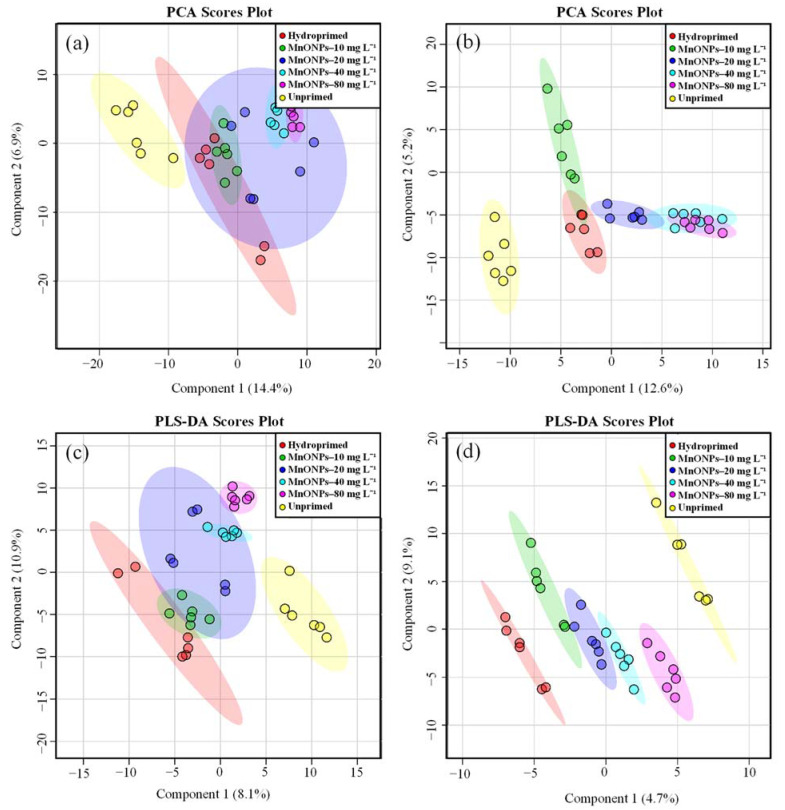
(**a**,**b**) indicate principal component analysis (PCA) scores plots of leaf tissue metabolomes of unprimed, hydroprimed, and MnO-NPs-primed seedlings of diploid and triploid watermelon varieties, respectively. (**c**,**d**) show the PLS-DA scores plots of leaf tissue metabolomes of diploid and triploid watermelon seedlings after different treatments.

**Figure 5 nanomaterials-11-01016-f005:**
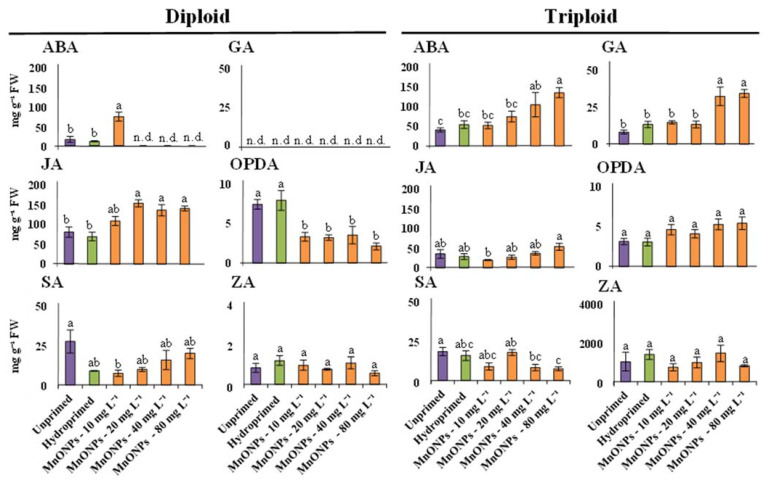
Leaf tissue hormonal profiles in unprimed, hydroprimed, and MnO-NPs-primed diploid and triploid watermelon seedlings. Letters show significant (*p* < 0.05) differences based on a post hoc Tukey’s test (ABA, abscisic acid; GA, gibberellic acid; JA, jasmonic acid; OPDA, 12-oxo phytodienoic acid; SA, salicylic acid; ZA, zeatin). Different letters indicate significant differences (*p* < 0.05) among treatments according to Tukey’s multiple-range test and n.d. represents not detected.

**Table 1 nanomaterials-11-01016-t001:** The levels of certain phenolic acids in leaf tissue samples of unprimed, hydroprimed, and MnO-NPs-primed diploid and triploid watermelon seedlings. Different letters indicate significance at *p* < 0.05.

Genotype	Treatment	Phenolic Acid Content (mg·g^−1^ of Fresh Weight)				
		4-Hydroxy-Benzoic Acid	Caffeic Acid	Phthalic Acid	Protocatechuic Acid	*Trans*-Cinnamic Acid
Diploid	Unprimed	1.54 ± 0.2 a	6.52 ± 1.1 a	8.70 ± 1.5 a	2.46 ± 0.9 a	42.1 ± 2.2 ab
	Hydroprimed	2.18 ± 0.4 a	5.08 ± 0.5 ab	6.17 ± 0.8 a	0.69 ± 0.1 a	47.9 ± 5.1 ab
	MnONPs—10 mg·L^−1^	2.05 ± 0.7 a	3.47 ± 0.3 ab	7.06 ± 0.7 a	0.83 ± 0.2 a	36.4 ± 4.6 b
	MnONPs—20 mg·L^−1^	1.71 ± 0.2 a	4.78 ± 0.6 ab	5.03 ± 0.9 a	0.52 ± 0.2 a	44.9 ± 10.0 b
	MnONPs—40 mg·L^−1^	2.08 ± 0.3 a	4.96 ± 0.7 ab	7.71 ± 1.4 a	0.53 ± 0.2 a	42.1 ± 6.0 ab
	MnONPs—80 mg·L^−1^	1.71 ± 0.1 a	2.70 ± 0.2 b	6.98 ± 1.3 a	n.d.	68.2 ± 12.8 a
Triploid	Unprimed	1.52 ± 0.1 a	3.57 ± 0.2 a	6.77 ± 1.0 a	0.72 ± 0.2 ab	31.7 ± 5.6 b
	Hydroprimed	1.53 ± 0.2 a	2.83 ± 0.2 a	10.36 ± 1.3 a	0.55 ± 0.3 ab	32.1 ± 3.2 b
	MnONPs—10 mg·L^−1^	1.43 ± 0.2 a	3.16 ± 0.3 a	7.08 ± 0.9 a	1.25 ± 0.2 a	43.0 ± 14.6 b
	MnONPs—20 mg·L^−1^	1.60 ± 0.1 a	3.46 ± 0.5 a	7.13 ± 1.6 a	1.01 ± 0.3 a	29.3 ± 3.9 b
	MnONPs—40 mg·L^−1^	1.81 ± 0.1 a	3.45 ± 0.3 a	7.89 ± 0.5 a	0.98 ± 0.2 a	24.9 ± 7.4 b
	MnONPs—80 mg·L^−1^	1.68 ± 0.3 a	3.11 ± 0.6 a	6.86 ± 1.5 a	n.d.	90.1 ± 5.9 a

## Data Availability

The data presented in this study are available on request from the corresponding author.

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
