# Peer review of "Manganese Oxide Nanoparticles as Safer Seed Priming Agent to Improve Chlorophyll and Antioxidant Profiles in Watermelon Seedlings"

_nanomaterials, 2021, doi:10.3390/nano11041016_

Round 1

Reviewer 1 Report

The paper entitled: Biosynthesized Manganese Oxide Nanoparticles as Safer Seed Priming Agent to Improve Chlorophylls and Antioxidant Profiles in Watermelon Seedlings, by: D. M. Kasote, J. H. J. Lee, G. K. Jayaprakasha, and B. S. Patil, deals with the preparation of manganese oxide nanoparticles by using onion bulb extracts as reducing agent in the reaction with potassium permanganate solution, then the structural and morphological characterization of prepared manganese oxide nanoparticles were performed by using several classic techniques in nanomaterials. Then, the toxicity of seed priming with the prepared manganese oxide nanoparticles in comparison to KMnO4 and Mn2O3 was studied, and the effect of different concentrations of the prepared manganese oxide nanoparticles in seed priming treatments was studied by using tests developed by the group. The authors claim that seed priming with their prepared manganese oxide nanoparticles had less phytotoxicity in comparison to KMnO4 and Mn2O3 in bulk form, and their prepared manganese oxide nanoparticles considerably modulate chlorophylls and antioxidant profiles, therefore they were safer than their bulk counterparts for their use in increasing crop productivity. The paper is well described, a few typing mistakes are found along the text, my only concern is that the paper is too specialized for general purpose and it is probably of interest only for agricultural researchers.

Reviewer 2 Report

The introduction grounds the research in the literature quite well.  The methods used to perform the study are clear and adequate and the experimental design used is well explained. The results presented are adequate, presented in a clear manner. The tables and figures used to show them are adequate.

I believe that it would be of interest to the readership of Nanomaterials, and would be suitable for publication after some minor revisions. 

Some minor concerns regarding:

Discussion: The authors need improve the discussion section focusing the importance of study and the relevance of this study.

Reviewer 3 Report

This research is very meaningful, but there are several problems that seriously affect the quality of the article.

1, The authors studied MnO nanoparticles, but why use KMnO4 and Mn2O3 bulk as a controls?

2, Why did the author use 25, 50 and 100 mg.L-1 concentrations before, and then changed them to 10, 20, 40 and 80 mg.L-1 concentrations?

3, Why are there no KMnO4 and Mn2O3 bulk treatments in Fig.3, Fig.4, Fig.5 and table 2?

4, All Figures are of poor quality, such as abscissa.

5, Many literatures about the effects of nanoparticles on Seed Germination and seedling are missing.

Round 2

Reviewer 3 Report

Authors have addressed all my comments.